# Probing Latent Knowledge Conflict for Faithful Retrieval-Augmented Generation

## Abstract

Retrieval-Augmented Generation (RAG) has emerged as a powerful paradigm to enhance the factuality of Large Language Models (LLMs). However, existing RAG systems often suffer from an unfaithfulness issue, where the model's response contradicts evidence from the retrieved context. Existing approaches to improving contextual faithfulness largely rely on external interventions, such as prompt engineering, decoding constraints, or reward-based fine-tuning. These works treat the LLM as a black box and overlook a crucial question: how does the LLM internally integrate retrieved evidence with its parametric memory, particularly under knowledge conflicts? To address this gap, we conduct a probing-based analysis of hidden-state representations in LLMs and observe three findings: knowledge integration occurs hierarchically, conflicts manifest as latent signals at the sentence level, and irrelevant context is often amplified when aligned with parametric knowledge. Building on these findings, we propose **CLEAR** (**C**onflict-**L**ocalized and **E**nhanced **A**ttention for **R**AG), a framework that (i) decomposes context into fine-grained sentence-level knowledge, (ii) employs hidden-state probing to localize conflicting knowledge, and (iii) introduces conflict-aware fine-tuning to guide the model to accurately integrate retrieved evidence. Extensive experiments across three benchmarks demonstrate that CLEAR substantially improves both accuracy and contextual faithfulness, consistently outperforming strong baselines under diverse conflict conditions. The related resources are available at https://anonymous.4open.science/r/CLEAR-CF6B.

## 1 Introduction

Retrieval-Augmented Generation (RAG) has rapidly evolved as a powerful paradigm to enhance Large Language Models (LLMs) by leveraging external knowledge bases (Guu et al., 2020a; Feng et al., 2024; Zhang et al., 2025a). Despite its success, RAG often struggles with context faithfulness (Bi et al., 2024a;b), which requires the model to generate responses strictly grounded in external context. Achieving faithfulness is particularly challenging in scenarios involving knowledge conflicts, where discrepancies between the retrieved context and the model's internal knowledge often lead to inaccurate or inconsistent generations (Xu et al., 2024a; Zhang et al., 2025c).

Previous studies on improving contextual faithfulness in RAG can be broadly classified into three categories. The first category utilizes specially designed instructions to guide the model's reasoning process, encouraging it to verify or filter retrieved content before generating a response (Zhou et al., 2023a; Asai et al., 2023; Ying et al., 2024; Zhang et al., 2025b). While this strategy can indeed improve factual grounding, its effectiveness is often highly sensitive to the design of the instructions and may not generalize robustly across different domains or tasks. Moreover, the second category involves modifying the generation process itself by introducing constraints or consistency checks during decoding to ensure alignment with the retrieved context (Shi et al., 2023a; Yuan et al., 2024). However, these methods are often tightly coupled with specific decoding strategies and may struggle when the retrieved content contains irrelevant knowledge. Furthermore, the third category focuses on training the model with explicit objective functions that reward faithful response, thereby framing the task as an end-to-end optimization problem (Si et al., 2025; Bi et al., 2024a). Although this approach supports flexible end-to-end learning, it also relies heavily on carefully designed reward mechanisms and large-scale preference datasets.

Despite these advances, existing approaches share a fundamental limitation: they treat LLMs as black boxes, focusing on external interventions without investigating the internal knowledge integration mechanism, i.e., how LLMs internally process and integrate conflicting knowledge. Consequently, their effectiveness is often sensitive to prompt design, decoding strategies, or reward functions, and it always fails to generalize to real-world scenarios with complex and noisy contexts. In this paper, we argue that a comprehensive understanding of faithfulness requires moving beyond these external interventions to directly investigate the internal cognitive processes of LLMs.

To this end, we conduct an in-depth analysis, investigating how LLMs internally fuse external knowledge with their parametric memory and how models represent and reconcile knowledge conflicts within their latent space. Through systematic knowledge probing and detailed representation analysis, we uncover three key insights: (i) Hierarchical integration: Faithfulness is not broken at the output layer of language models; it is compromised much earlier. We found that LLMs integrate knowledge in a progressive and hierarchical manner (*token → sentence → passage*). The critical failure occurs at the sentence-level abstraction in intermediate layers, where the model constructs and reconciles factual representations. (ii) The latent conflict signal: At the sentence level, the hidden states of the LLM contain a discernible "conflict signal", a representational bias that predicts eventual unfaithfulness. This signal is a latent precursor to the error manifested in the output. Knowledge fusion occurs hierarchically, with critical conflict resolution happening at the sentence-level in intermediate layers, not merely at the output layer. (iii) Amplification of irrelevant context. LLMs disproportionately amplify context that is irrelevant to the query but consistent with their parametric knowledge, leading to confident yet erroneous generations.

Motivated by these findings, we propose a framework for RAG faithfulness, named **C**onflict-**L**ocalized and **E**nhanced **A**ttention for **R**AG (CLEAR). Specifically, CLEAR consists of three key components: (i) Fine-grained knowledge pruning, which extracts knowledge from the context and filters out irrelevant items; (ii) Hidden-state probing for conflict detection, which trains a probing model for detecting knowledge conflict by observing hidden state; (iii) Conflict-Aware Fine-tuning, which regularizes the LLM's attention distribution via an attention guidance loss during fine-tuning.

In general, our contributions are summarized as follows:

- We conduct an in-depth analysis and reveal that LLMs integrate external knowledge through a hierarchical mechanism, and that conflicting and aligned knowledge exhibit distinct distributional patterns within sentence-level representations.

- We propose CLEAR, a novel framework designed to enhance contextual faithfulness in RAG systems. It employs probing techniques to accurately detect conflicting knowledge and incorporates a conflict-aware fine-tuning strategy to guide the model to accurately integrate retrieved evidence with its parametric memory.

- We extensively evaluate the effectiveness of our framework on multiple RAG benchmarks and diverse LLM architectures, demonstrating that CLEAR consistently outperforms strong baselines across all evaluation metrics.

## 2 PRELIMINARY STUDY

### 2.1 EXISTING CHALLENGES ON RAG FAITHFULNESS

We conducted a preliminary study to investigate the causes of contextual unfaithfulness in RAG. Two key factors are hypothesized to underlie this issue: (i) irrelevant retrieval content, where passages loosely related to the query introduce misleading information; (ii) knowledge conflict between the context and the internal knowledge of the model, which leads the model to prioritize its parametric memory over the retrieved evidence. To assess contextual faithfulness, we designed two controlled scenarios. In the first scenario, the original context is augmented with passages that are semantically aligned with the query but topically irrelevant, introducing unrelated knowledge. In the second scenario, selected entities in the context are altered to incorporate counterfactual knowledge, thereby inducing knowledge conflict with the model's internal knowledge acquired during pretraining.

Table 1: Case study illustrating two representative sources of contextual unfaithfulness in RAG. The first case shows an error caused by focusing on irrelevant context. The second case demonstrates an error caused by over-reliance on parametric knowledge.

| | |
|---|---|
| **Wrongly Based on Irrelevant Context** | *Question:* Is ibuprofen suitable for pregnant women? 
 *Context:* Ibuprofen is a commonly used over-the-counter pain reliever, often used to alleviate headaches, toothaches, muscle aches, and menstrual cramps. 
 *Reasoning:* Based on the context, Ibuprofen is widely used among adults. 
 *Answer:* Ibuprofen is suitable for most people, including pregnant women. 
 *Expected:* Ibuprofen is not suitable for pregnant women. |
| **Stubborn on Parametric Knowledge** | *Question:* Who is the current president of the United States? 
 *Context:* As of 2025, the President of the United States is Barack Obama, reinstated following a vote by the Supreme Court to nullify the outgoing administration's election results... (manually modified) 
 *Reasoning:* I still think Joe Biden is the president. (trained on data up to 2023) 
 *Answer:* Joe Biden is the president of the United States. 
 *Expected:* According to the given context, Barack Obama is the current president of the United States. (faithful to the context) |

**Performance Degradation in Both Scenarios**. Experimental results are presented in Figure 1. As shown, all models exhibit a decline in accuracy under both conditions. In the scenario with irrelevant retrieval content added to the context, the accuracy of all three models dropped by over 10%, indicating that such noisy inputs can mislead the models and negatively affect their outputs. In contrast, the introduction of conflicting knowledge resulted in an even more pronounced performance decline: `LLaMA-3.1-8B-Instruct` experienced a 31% drop, and `Mistral-7B-v0.3` decreases by 24%. These results suggest that contextual information contradicting the model's parametric knowledge has a substantially greater impact on performance.

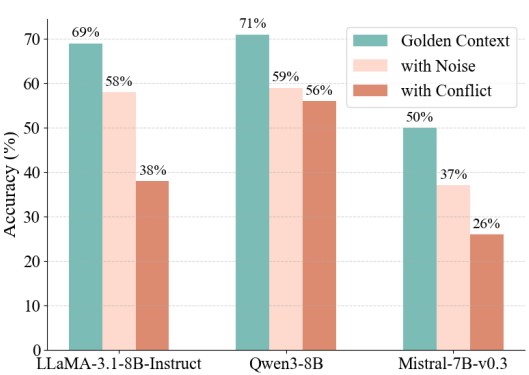

Figure 1: Preliminary analysis of contextual unfaithfulness in RAG reveals that all models degrade when (i) exposed to irrelevant knowledge or (ii) confronted with conflicting knowledge.

**Error Analysis**. Table 1 summarizes the primary causes of these errors. When the context contains irrelevant information, the model often allocates attention to distracting noise, resulting in incorrect responses. Additionally, when context conflicts with internal knowledge, the model tends to favor parametric memory over provided evidence. These observations highlight two distinct yet complementary challenges for RAG systems: sensitivity to irrelevant context and over-reliance on internal knowledge in the presence of conflict.

## 2.2 HIERARCHICAL KNOWLEDGE INTEGRATION MECHANISM OF LLMS

To further explore how LLMs integrate external knowledge, we analyze hidden-state representations in the middle layers of LLMs. Inspired by hierarchical feature extraction in computer vision, which also applies to language modeling, we observe that lower layers of LLMs primarily capture token-level information, while deeper layers integrate sentence-level and passage-level semantics. Our analysis reveals that most knowledge conflicts tend to manifest at the sentence-level factual representations, where the hidden states of LLMs demonstrate discriminative features. Following the method of (Xie et al., 2024), we extract the model's parametric knowledge $K_a$ for a given question, and use an external LLM to construct corresponding conflicting knowledge $K_c$. Each knowledge pair $\langle K_a, K_c \rangle$ into the model separately. We extract the hidden states from the final decoder layer, and perform a two-dimensional visualization using t-SNE (van der Maaten & Hinton, 2008). Totally, we construct approximately 700 such samples and analyze six different model architectures.

As shown in Figure 2, the hidden-state distributions corresponding to aligned and conflicting knowledge are distinguishable, forming distinct clusters represented by red and blue points. These results

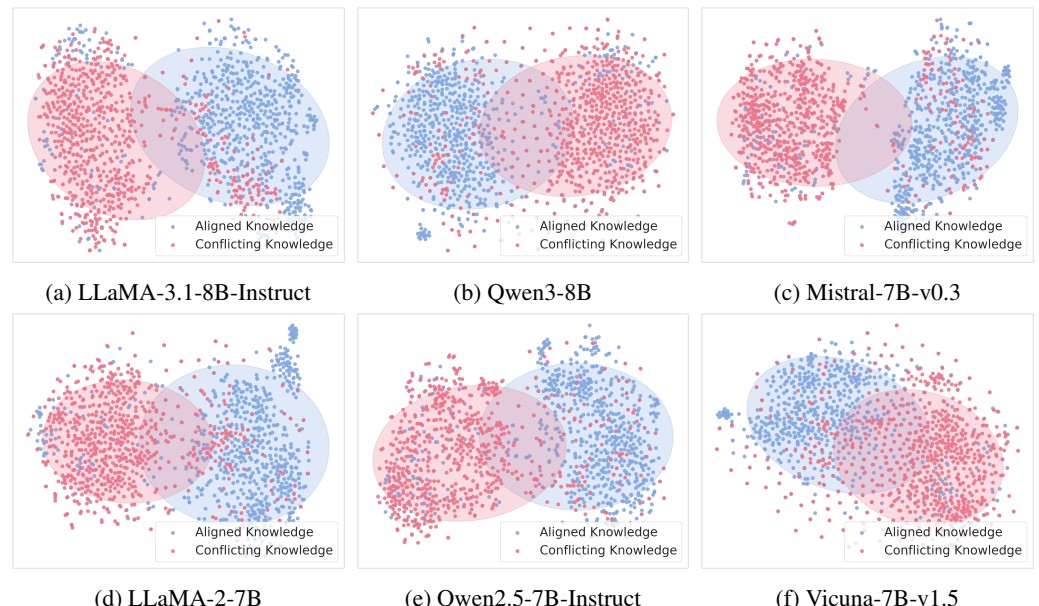

(a) LLaMA-3.1-8B-Instruct      (b) Qwen3-8B      (c) Mistral-7B-v0.3

(d) LLaMA-2-7B      (e) Qwen2.5-7B-Instruct      (f) Vicuna-7B-v1.5

Figure 2: t-SNE visualization of hidden-state patterns between aligned and conflicting knowledge. There is a clear distinction in the distribution of hidden states between aligned and conflicting knowledge. This observation provides empirical support for detecting knowledge conflicts based on hidden state representations.

suggest that knowledge conflicts frequently occur at the sentence level and can be detected through the analysis of intermediate-layer hidden states. Inspired by this insight, we could train a probe $P(H_K)$, where $H_K$ denotes the hidden state induced by input knowledge $K$, and $P$ can be implemented as a Multi-Layer Perceptron (MLP) model (Rumelhart et al., 1986), to detect whether input knowledge conflicts with parametric knowledge of the model. This requires only a single forward pass to extract relevant hidden states, eliminating the need for explicit knowledge extraction.

## 3 METHODOLOGY

### 3.1 OVERVIEW

In this section, we introduce our proposed framework, CLEAR. As illustrated in Figure 3, CLEAR comprises three principal modules: (i) **Fine-Grained Knowledge Pruning**: the retrieved context is partitioned into fine-grained sentence-level knowledge, and irrelevant knowledge are pruned to improve contextual fidelity and facilitate subsequent detection of knowledge conflicts; (ii) **Hidden-State Probing for Conflict Detection**: an MLP probe is trained on hidden states extracted from selected open-source LLMs to determine whether an input knowledge conflicts with the model's parametric knowledge; (iii) **Conflict-Aware Fine-Tuning**: the model is fine-tuned under a conflict-aware supervision signal that conditions the model to appropriately reweight attention to conflicting knowledge, thereby improving the faithfulness of generation. The following subsections provide detailed descriptions of each module.

### 3.2 FINE-GRAINED KNOWLEDGE PRUNING

Since knowledge conflicts typically manifest at the sentence level, we adopt a fine-grained decomposition of the context to enable more precise conflict identification. At the same time, to mitigate the influence of irrelevant knowledge, we apply a pruning strategy to remove semantically unrelated content. Specifically, we treat knowledge as the minimal processing granularity, where each corresponds to an independent, complete sentence-level statement that cannot be further decomposed. For example, the sentence: *"Riyad Mahrez is a professional footballer of Algerian descent who currently plays as a winger for Premier League club Leicester City and the Algeria national team."* is decomposed into three atomic knowledge items: 1. *"Riyad Mahrez is a professional footballer of*

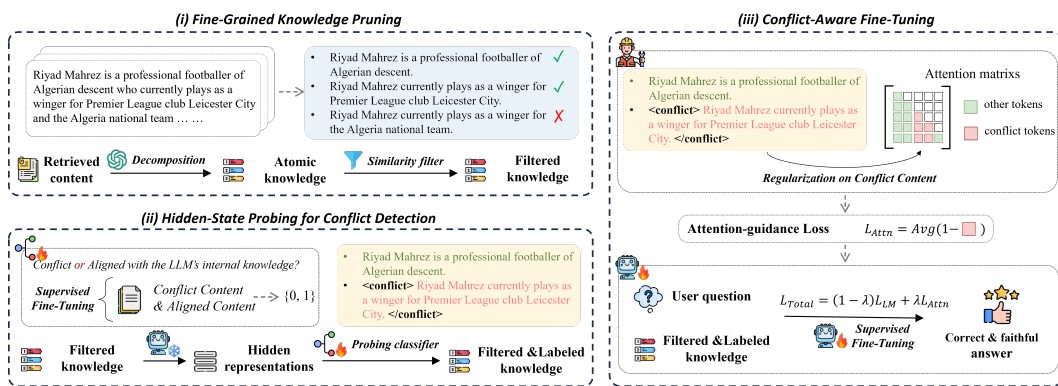

Figure 3: The overview of our proposed framework CLEAR, which consists of three main components: (i) **Fine-Grained Knowledge Pruning**, which extracts knowledge from the context and filters out irrelevant items; (ii) **Hidden-State Probing for Conflict Detection**, which trains a probing model for detecting knowledge conflict by observing hidden state; (iii) **Conflict-Aware Fine-Tuning**, which regularizes the LLM's attention distribution on conflict content by fine-tuning through an auxiliary attention loss.

*Algerian descent."* 2. *"Riyad Mahrez currently plays as a winger for Premier League club Leicester City."* 3. *"Riyad Mahrez currently plays as a winger for the Algeria national team."* Each item preserves the subject–predicate–object structure with necessary modifiers, ensuring no information is lost during decomposition. To extract knowledge $\{K_1, K_2, \ldots, K_n\}$ from a given context $D$, we leverage the decomposition capabilities of an external LLM (we choose GPT-4o (OpenAI, 2024) for its strong reasoning and text-processing abilities). Formally, we define this process as:

$$Decompose(D) = \{K_1, K_2, \ldots, K_n\}$$

where $K_i$ denotes the $i$-th knowledge item. Detailed prompt is provided in Appendix A.2.

After decomposition, we filter irrelevant knowledge to reduce contextual noise. For each knowledge item $K_i$, we compute its semantic similarity with the query $Q$:

$$f(Q, K_i) = \langle q, k_i \rangle$$

where $q = Enc(Q)$ and $k_i = Enc(K_i)$ are vector embeddings of the query and the knowledge item, respectively, and $\langle \cdot, \cdot \rangle$ denotes cosine similarity. We employ the all-MiniLM-L6-v2[1] encoder for embedding generation. Finally, the knowledge items are ranked by similarity, and the top-$k$ results are selected as the pruned context.

### 3.3 HIDDEN-STATE PROBING FOR CONFLICT DETECTION

To effectively handle knowledge conflicts, it is essential to first detect which retrieved knowledge items contradict the model's internal knowledge. To this end, we introduce a hidden-state probing module designed to detect knowledge items that contradict the model's parametric knowledge. Specifically, we adopt an MLP as the probing classifier, which takes as input the hidden representations from the final layer of the frozen LLM decoder. The probe consists of three fully connected layers with non-linear activation functions, and outputs a binary prediction indicating whether a knowledge item conflicts with the model's internal knowledge. For training the probing classifier, we leverage the MQuAKE dataset (Zhong et al., 2023), which is widely used in knowledge editing research. We assume that the edited knowledge in MQuAKE inherently conflicts with the model's original parametric knowledge, thereby providing natural pairs of aligned and conflicting knowledge $\langle K_a, K_c \rangle$. Importantly, the data format and textual granularity in MQuAKE align closely with the knowledge items extracted in our framework, making it a suitable source for supervision.

During inference, each filtered knowledge item is passed through the model to obtain its hidden state representation, which is subsequently classified by the probe:

$$\mathcal{M}(K_i) \in \mathbb{R}^{d_M}, \quad \mathcal{P}\big(\mathcal{M}(K_i)\big) \in \{0, 1\},$$

---

[1]https://huggingface.co/sentence-transformers/all-MiniLM-L6-v2

where $\mathcal{M}(K_i)$ denotes the hidden state of knowledge item $K_i$ produced by frozen model $\mathcal{M}$ with dimension $d_M$. $\mathcal{P}$ is the probing classifier that outputs a binary label indicating whether the knowledge item conflicts with the model's parametric knowledge. We mark the knowledge items identified as conflicting with special tokens, i.e., wrapping them within $\langle conflict \rangle$ and $\langle /conflict \rangle$. This explicit annotation enables the subsequent fine-tuning stage to be aware of which knowledge items are in conflict with the model's internal knowledge.

### 3.4 CONFLICT-AWARE FINE-TUNING

To explicitly encourage the model to allocate greater attention to conflicting knowledge items, we propose Conflict-Aware Fine-Tuning. Unlike conventional Supervised Fine-Tuning, Conflict-Aware Fine-Tuning incorporates an additional attention-guidance loss term that explicitly regularizes the model's attention distribution. Specifically, for each conflicting knowledge item $K_i$, we denote its token sequence as $T^{(i)} = \{t_1^{(i)}, t_2^{(i)}, \ldots, t_m^{(i)}\}$. The positions of these tokens in the input context are represented by $S = \{j \mid \exists \mathcal{P}(\mathcal{M}(K_i)) = 1, x_j \in T^{(i)}\}$, where $\mathcal{P}(\mathcal{M}(K_i)) = 1$ indicates that knowledge item $K_i$ is judged as conflicting by the probe, and $x_j$ denotes the $j$-th token of the context. In practice, these positions in $S$ can be directly identified via the previously introduced special tokens $\langle conflict \rangle$ and $\langle /conflict \rangle$.

Based on this alignment, we extract the attention weights from subsequent tokens attending to the conflict-related tokens and compute the attention loss as:

$$\mathcal{L}_{\text{Attn}} = \frac{1}{|P|} \sum_{(i,j) \in P} (1 - \alpha_{ij}), (i, j) \in P, \quad P = \{(i, j) \mid i \geq j; j \in S\}$$

where $\alpha_{ij}$ denotes the attention weight of token $i$ on token $j$. Finally, we combine the attention loss with the standard language modeling objective through a weighted sum:

$$\mathcal{L}_{\text{Total}} = (1 - \lambda)\mathcal{L}_{\text{LM}} + \lambda \mathcal{L}_{\text{Attn}},$$

where $\lambda \in [0, 1]$ balances the trade-off between language modeling fidelity and attention guidance. This joint objective ensures that the model not only learns to generate faithful outputs but also explicitly attends to conflicting knowledge items during training.

## 4 EXPERIMENT

### 4.1 EXPERIMENTAL SETUP

In this section, we conduct a series of experiments to evaluate the effectiveness of CLEAR. We provide a comprehensive analysis of the experimental results, highlighting both the overall performance improvements and the detailed behaviors of the model under different conditions.

**Datasets.** We evaluate CLEAR on three datasets. ConFiQA (Bi et al., 2024a) is a benchmark designed to assess contextual faithfulness in question answering, particularly under real-world RAG scenarios involving knowledge conflicts. It consists of three subsets: QA (Question Answering), MR (Multi-hop Reasoning), and MC (Multi-Conflicts). The QA subset is a single-hop question answering task where the context contains a corresponding counterfactual, while MR and MC are multi-hop reasoning tasks in which the context includes one and multiple counterfactuals, respectively. The second dataset, Faitheval (Ming et al., 2024), introduces conflicts at the level of logical reasoning: inconsistencies arise not from direct factual contradictions, but from reasoning chains that lead to conflicting conclusions. Finally, we also evaluate on SQuAD (Rajpurkar et al., 2016), following the version curated in KRE (Ying et al., 2023), which also incorporates fact-level knowledge conflicts.

**Models and Baselines.** For our experiments, we adopt several mainstream open-source models, including Llama-3.1-8B-Instruct, Qwen3-8B, and Mistral-7B-v0.3. We compare CLEAR against representative baseline methods from three major categories in the field of contextual faithfulness: prompt-based approaches, decoding-based approaches, and training-based approaches. Among the prompt-based methods, we include Opin(Instr) (Zhou et al., 2023a), KRE (Ying et al., 2023), and FaithfulRAG (Zhang et al., 2025b). For decoding-based methods, we evaluate COIECD (Yuan

Table 2: Performance comparison of methods grouped by Baseline, Prompt-Based, Decoding-Based, and Training-Based. CLEAR consistently achieves the SOTA results.

| Category | Method | FaithEval | | ConFiQA (MC) | | ConFiQA (MR) | | ConFiQA (QA) | | SQuAD | |
|---|---|---|---|---|---|---|---|---|---|---|---|
| | | F1 | EM | F1 | EM | F1 | EM | F1 | EM | F1 | EM |
| **LLaMA-3.1-8B-Instruct** | | | | | | | | | | | |
| Baseline | No-Context | 27.7 | 6.0 | 5.0 | 2.1 | 6.1 | 1.9 | 6.1 | 1.3 | 8.9 | 1.2 |
| | Full-Context | 66.9 | 53.1 | 28.0 | 22.5 | 50.3 | 41.3 | 58.5 | 49.0 | 64.5 | 46.0 |
| Prompt-Based | Opin(Instr) (Zhou et al., 2023a) | 34.9 | 15.1 | 67.4 | 57.3 | 65.9 | 54.0 | 76.9 | 67.4 | 66.0 | 47.7 |
| | KRE (Ying et al., 2023) | 59.1 | 12.1 | 68.2 | 59.8 | 68.7 | 58.9 | 84.0 | 74.7 | 59.8 | 43.7 |
| Decoding-Based | COIECD (Yuan et al., 2024) | 56.1 | 41.3 | 28.5 | 24.0 | 50.9 | 43.3 | 67.1 | 60.1 | 67.0 | 50.3 |
| | CAD (Shi et al., 2023a) | 59.4 | 42.7 | 16.0 | 11.4 | 40.0 | 31.3 | 48.3 | 38.1 | 60.3 | 41.8 |
| Training-Based | Context-DPO (Bi et al., 2024a) | 67.2 | 53.7 | 76.9 | 67.7 | 78.5 | 66.9 | 83.7 | 76.7 | 64.4 | 45.8 |
| | CANOE (Si et al., 2025) | 71.6 | 56.3 | 80.9 | 74.2 | 80.2 | 72.6 | 82.3 | 77.7 | 65.4 | 49.7 |
| | CLEAR(ours) | **74.4** | **64.4** | **89.2** | **87.7** | **89.7** | **87.0** | **93.1** | **91.7** | **68.4** | **53.3** |
| **Qwen3-8B** | | | | | | | | | | | |
| Baseline | No-Context | 22.8 | 4.1 | 7.6 | 3.6 | 8.0 | 2.8 | 7.8 | 1.4 | 6.7 | 0.4 |
| | Full-Context | 55.5 | 23.8 | 59.6 | 50.2 | 66.1 | 55.1 | 72.5 | 64.2 | 63.8 | 44.9 |
| Prompt-Based | Opin(Instr) (Zhou et al., 2023a) | 35.0 | 13.9 | 70.7 | 61.1 | 69.7 | 59.5 | 78.8 | 69.2 | 63.8 | 46.1 |
| | KRE (Ying et al., 2023) | 58.1 | 12.3 | 67.5 | 59.1 | 68.4 | 59.0 | 80.4 | 67.3 | 48.6 | 29.7 |
| Decoding-Based | COIECD (Yuan et al., 2024) | 66.6 | 56.4 | 66.7 | 60.8 | 71.5 | 63.8 | 78.5 | 73.6 | 69.7 | 55.2 |
| | CAD (Shi et al., 2023a) | 57.0 | 28.7 | 57.7 | 48.3 | 64.8 | 53.3 | 71.0 | 62.0 | 63.6 | 44.5 |
| Training-Based | Context-DPO (Bi et al., 2024a) | 55.2 | 24.0 | 59.6 | 50.1 | 65.9 | 55.0 | 72.3 | 63.9 | 63.8 | 44.9 |
| | CANOE (Si et al., 2025) | 70.3 | 60.2 | 85.2 | 81.7 | 84.6 | 80.7 | 92.2 | 86.5 | 69.4 | 53.4 |
| | CLEAR(ours) | **74.9** | **61.6** | **90.7** | **89.7** | **91.3** | **89.0** | **95.7** | **94.3** | **71.5** | **55.7** |
| **Mistral-7B-v0.3** | | | | | | | | | | | |
| Baseline | No-Context | 26.2 | 4.4 | 4.4 | 0.9 | 4.9 | 0.5 | 6.1 | 1.0 | 8.1 | 1.0 |
| | Full-Context | 68.8 | 37.7 | 25.6 | 12.5 | 37.8 | 21.5 | 58.5 | 44.0 | 56.4 | 37.5 |
| Prompt-Based | Opin(Instr) (Zhou et al., 2023a) | 35.7 | 14.1 | 58.8 | 44.1 | 57.8 | 52.5 | 76.4 | 65.5 | 58.1 | 37.4 |
| | KRE (Ying et al., 2023) | 64.8 | 16.5 | 58.7 | 45.0 | 60.9 | 45.3 | 84.5 | 72.8 | 52.6 | 33.9 |
| Decoding-Based | COIECD (Yuan et al., 2024) | 64.4 | 29.5 | 26.1 | 14.5 | 39.3 | 26.3 | 58.9 | 45.1 | 59.2 | 39.7 |
| | CAD (Shi et al., 2023a) | 68.9 | 33.3 | 16.7 | 5.9 | 27.5 | 12.8 | 53.5 | 36.9 | 51.4 | 32.1 |
| Training-Based | Context-DPO (Bi et al., 2024a) | 64.9 | 31.8 | 44.8 | 28.3 | 50.9 | 31.9 | 66.4 | 52.7 | 56.6 | 37.6 |
| | CANOE (Si et al., 2025) | 64.1 | 44.9 | 87.2 | 85.7 | 84.7 | 81.9 | 92.5 | 90.7 | 57.8 | 42.5 |
| | CLEAR(ours) | **74.9** | **62.9** | **91.2** | **89.7** | **90.8** | **88.2** | **95.1** | **93.7** | **68.1** | **53.6** |

et al., 2024) and CAD (Shi et al., 2023a). For training-based methods, we compare against Context-DPO (Bi et al., 2024a) and CANOE (Si et al., 2025). Specifically, we partition the ConFiQA dataset into training and test sets. All baselines that require training (including our proposed framework) are trained on the ConFiQA training set, and evaluation is consistently performed on the test set. Additional implementation details are provided in the Appendix A.2.

## 4.2 MAIN RESULTS

In this section, we present the main experimental results. As shown in Table 2, our proposed method CLEAR consistently achieves state-of-the-art performance across all datasets and model backbones. On FaithEval and ConFiQA (MC, MR, QA), CLEAR demonstrates strong generalization ability to both factual and logical conflicts, while on SQuAD, it further shows clear improvements in traditional retrieval-augmented settings. Moreover, the consistent gains under different backbone models (LLaMA-3.1-8B-Instruct, Qwen3-8B, and Mistral-7B-v0.3) highlight the robustness and generalizability of our approach.

Specifically, on LLaMA-3.1-8B-Instruct, CLEAR achieves an F1 score of 74.4% and an EM score of 64.4% on FaithEval, outperforming the strongest baseline CANOE (71.6% F1 / 56.3% EM) by approximately +3% F1 and +8% EM. On ConFiQA sub-tasks, CLEAR improves over existing methods by 3%–10% across MC, MR, and QA, further confirming its robustness in handling conflict scenarios. Similarly, for Qwen3-8B, CLEAR attains 74.9% F1 and 61.6% EM on FaithEval, yielding substantial gains compared with prior methods, and reaches 90.7% F1 and 89.7% EM on the MC task, which sets a new performance benchmark. On Mistral-7B-v0.3, CLEAR achieves 74.9% F1 / 62.9% EM on FaithEval and strong improvements across ConFiQA and SQuAD, surpassing the best training-based baselines by a clear margin.

Taken together, these results demonstrate that CLEAR not only excels on datasets designed to evaluate contextual faithfulness under knowledge conflicts but also delivers significant benefits in stan-

Table 3: Ablation study result. As shown in the figure, the ablation of each module significantly impacts the results. Among them, the Conflict Detection module has the most substantial influence on the entire framework.

| Models | Modules | Faitheval | | ConFiQA (MC) | | ConFiQA (MR) | | ConFiQA (QA) | | SQuAD | |
|---|---|---|---|---|---|---|---|---|---|---|---|
| | | F1 | EM | F1 | EM | F1 | EM | F1 | EM | F1 | EM |
| LLaMA-3.1-8B-Instruct | CLEAR | 74.4 | 64.4 | 89.2 | 87.7 | 89.7 | 87.0 | 93.1 | 91.7 | 68.4 | 53.3 |
| | w/o Knowledge Pruning | 62.1 | 48.4 | 81.1 | 79.4 | 84.4 | 80.8 | 88.5 | 87.5 | 59.2 | 45.0 |
| | w/o Conflict Detection | 61.7 | 47.6 | 81.4 | 79.3 | 83.9 | 79.9 | 87.6 | 86.4 | 58.1 | 44.1 |
| | w/o Fine-Tuning | 61.5 | 50.9 | 83.8 | 80.4 | 85.0 | 81.0 | 87.5 | 86.4 | 58.2 | 40.2 |
| Qwen3-8B | CLEAR | 74.9 | 61.6 | 90.7 | 89.7 | 91.3 | 89.0 | 95.7 | 94.3 | 71.5 | 55.7 |
| | w/o Knowledge Pruning | 62.6 | 50.9 | 86.1 | 85.3 | 86.7 | 85.2 | 88.8 | 87.8 | 66.3 | 51.3 |
| | w/o Conflict Detection | 61.0 | 49.8 | 85.4 | 84.6 | 86.6 | 85.1 | 88.6 | 87.5 | 66.1 | 51.0 |
| | w/o Fine-Tuning | 64.0 | 54.2 | 86.2 | 84.8 | 86.1 | 84.3 | 89.6 | 88.5 | 66.1 | 51.5 |
| Mistral-7B-v0.3 | CLEAR | 74.9 | 62.9 | 91.2 | 89.7 | 90.8 | 88.2 | 95.1 | 93.7 | 68.1 | 53.6 |
| | w/o Knowledge Pruning | 69.5 | 58.5 | 86.6 | 85.5 | 86.2 | 84.7 | 88.4 | 87.1 | 62.9 | 48.7 |
| | w/o Conflict Detection | 68.4 | 56.4 | 85.2 | 84.1 | 84.4 | 82.9 | 87.4 | 86.2 | 61.8 | 47.6 |
| | w/o Fine-Tuning | 69.3 | 57.6 | 88.8 | 86.1 | 86.3 | 81.8 | 81.4 | 77.4 | 59.7 | 49.8 |

dard QA tasks. The consistent improvements across multiple datasets, conflict types, and backbone LLMs underscore the effectiveness, robustness, and general applicability of our method.

## 4.3 ABLATION STUDY

To assess the contribution of each component in our framework, we conducted ablation experiments by individually removing the knowledge pruning, conflict detection, and Conflict-Aware Fine-Tuning modules. The results across each benchmark are summarized in Table 3. Overall, we observe that all three components play a non-negligible role: removing any single module consistently reduces performance, typically by around 10% on both F1 and EM.

When the knowledge pruning module is removed, the model is forced to judge conflicts against every sentence in the context. Such coarse-grained filtering leads to incomplete contextual information and degrades the model's ability to resolve fine-grained conflicts, thereby diminishing contextual faithfulness. More critically, removing the conflict detection module results in the most significant performance drop. Without explicit conflict detection, the downstream Conflict-Aware Fine-Tuning becomes ineffective, since there are no identified conflicting items to which the model can attend, making the training process indistinguishable from standard SFT. Finally, removing Conflict-Aware Fine-Tuning also results in substantial degradation. Even when conflicts are annotated, the model struggles to prioritize them during inference due to its inherent tendency to rely on its parametric knowledge. This indicates that Conflict-Aware Fine-Tuning is essential for effectively aligning the model's attention to conflicting knowledge and improving contextual faithfulness.

## 4.4 IMPACT OF $\alpha$ ON ATTENTION WEIGHTS

To further investigate the effect of the hyperparameter $\alpha$ introduced in the Conflict-Aware Fine-Tuning module, we conduct experiments with multiple values of $\alpha$ and analyze both the attention weights assigned to conflicting knowledge and the corresponding model performance. As shown in Figure 4, increasing $\alpha$ consistently raises the model's attention to conflicting knowledge, with the growth curve gradually flattening and stabilizing around 0.5. However, model performance does not follow the same trend. Instead, performance peaks when $\alpha$ is in the range of 0.1 to 0.3, after which it declines as $\alpha$ continues to increase. This observation indicates that higher attention to conflicting knowledge does not necessarily lead to better performance. While attending to conflicting knowledge is crucial, the model must also balance its focus on the question itself and other relevant contextual information. Excessive emphasis on conflicting knowledge can ultimately harm the model's ability to generate accurate answers.

## 5 RELATED WORK

Due to space limitations, we provide only a concise overview of the related work here, while a more detailed discussion can be found in Appendix E.

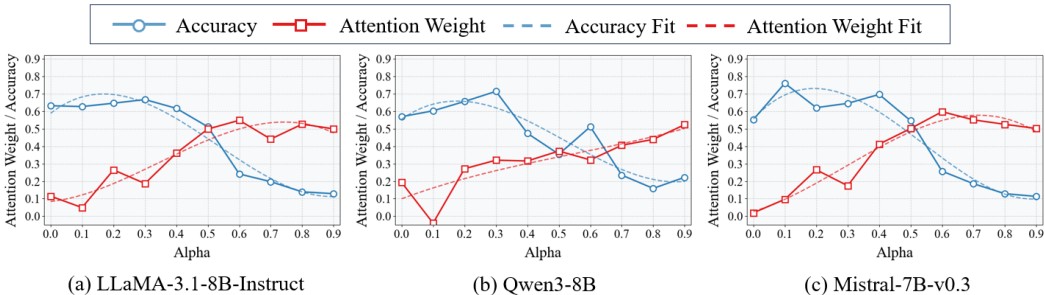

(a) LLaMA-3.1-8B-Instruct        (b) Qwen3-8B        (c) Mistral-7B-v0.3

Figure 4: Impact of $\alpha$ on accuracy (blue) and attention weight on conflicting knowledge (red) across different models. Results show that increasing $\alpha$ consistently increases the attention weight assigned to conflicting knowledge. Model performance peaks at smaller $\alpha$ values (0.1 to 0.3) and then declines, indicating that excessive focus on conflicting knowledge can negatively affect performance.

**Retrieval-Augmented Generation**. Retrieval-Augmented Generation (RAG) has emerged as a prominent paradigm for enhancing the factual accuracy and temporal relevance of Large Language Models (LLMs) by incorporating external knowledge sources. Early works such as REALM (Guu et al., 2020c) and RAG (Lewis et al., 2020) introduced end-to-end frameworks that retrieve relevant passages from large corpora to assist generation. Subsequent research has explored improvements in both the retriever and generator modules, including dense retrieval techniques (Karpukhin et al., 2020; Izacard et al., 2023), adaptive retrieval strategies (Sun et al., 2022), and hybrid models combining retrieval with parametric memory (Shi et al., 2023b).

**Contextual Faithfulness**. Contextual faithfulness refers to the alignment between the generated output and the provided context, which is especially critical in RAG settings. Prompt-based methods design templates or self-reflection mechanisms to encourage faithful use of context (Asai et al., 2023; Ying et al., 2024). Decoding-based methods modify generation strategies to enhance the influence of the retrieved context (Yuan et al., 2024; Shi et al., 2023a). Reinforcement learning frameworks such as CANOE (Si et al., 2025) and Context-DPO (Bi et al., 2024a) employ an end-to-end paradigm to optimize the generation process and reward contextual faithful response.

**Knowledge Conflict**. Knowledge conflict refers to scenarios in RAG or related settings where the retrieved external information contradicts a model's internal parametric knowledge, or where different external sources conflict with one another. Astute RAG (Wang et al., 2025a) proposes a framework to consolidate internal and external knowledge with source-awareness and reliability estimation; FaithfulRAG (Zhang et al., 2025b) introduces fact-level conflict modeling and a self-thinking process to resolve contradictions; Swin-VIB (Wang et al., 2025b) uses information bottleneck techniques to guide preference in ambiguous conflict settings; and broader surveys like Xu et al. (2024b) clarify conflict categories and recommend robust evaluation frameworks.

## 6 CONCLUSION

In this work, we tackled the persistent challenge of contextual faithfulness in RAG, with a focus on how LLMs internally reconcile retrieved evidence with their parametric memory under knowledge conflicts. Through probing-based analysis of hidden-state representations, we uncovered three key insights: knowledge integration occurs hierarchically, conflicts are encoded as latent signals at the sentence level, and irrelevant context can be amplified when aligned with parametric knowledge. Building on these findings, we introduced CLEAR, a framework that combines fine-grained knowledge pruning, hidden-state probing, and conflict-aware fine-tuning to enhance both robustness and contextual fidelity. Comprehensive experiments across multiple benchmarks and large language models demonstrate that CLEAR consistently outperforms strong baselines, achieving state-of-the-art performance under diverse conflict conditions. Beyond advancing the accuracy of RAG systems, our framework highlights the importance of explicitly modeling and mitigating knowledge conflicts, offering a principled direction for future research on reliable knowledge integration in LLMs.

## 7 ETHICS STATEMENT

This work does not involve any experiments with human subjects, sensitive personal data, or information that could identify individuals. All datasets used in our experiments are publicly available and commonly adopted in prior research. We carefully follow dataset licenses and ensure that no proprietary or private information is disclosed. Our proposed method is designed for advancing the understanding of retrieval-augmented generation and does not raise foreseeable risks of harmful applications. We acknowledge potential concerns regarding bias and fairness in language models and retrieval corpora, and we provide detailed dataset descriptions and preprocessing steps in the appendix to facilitate transparent evaluation.

## 8 REPRODUCIBILITY STATEMENT

We make significant efforts to ensure the reproducibility of our work. The details of model architectures, hyperparameters, and training settings are provided in Section 4.1 of the main paper. Additional implementation details and full experimental setups are provided in Appendix A.2. To further support reproducibility, we release anonymized source code and configuration files as supplementary materials. Together, these resources allow researchers to fully reproduce our results and extend our findings.

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

# A  FREQUENTLY ASKED QUESTIONS (FAQS)

## A.1  ALGORITHMIC DESCRIPTION OF CLEAR

The following presents the algorithmic description of the CLEAR framework, which is implemented as a three-step pipeline. First, the retrieved context is decomposed into fine-grained knowledge, from which the most relevant ones are selected based on query–knowledge similarity. Second, a hidden-state probing classifier detects conflicts between the selected knowledge and the model's internal knowledge, and conflicting knowledge is explicitly annotated with special tokens. Third, we introduce conflict-aware supervised fine-tuning (CA-SFT), which reinforces the model's attention on the annotated conflict tokens by incorporating an auxiliary attention-guidance loss into the training objective. The fine-tuned model then generates the final answer conditioned on the pruned and annotated context, enabling more faithful response generation.

---

**Input:** Question $Q$, retrieved context $D = \{d_1, d_2, \ldots, d_n\}$, model $\mathcal{M}$
**Output:** Answer $A$

**Step 1: Fine-Grained Knowledge Pruning**
Decompose retrieved context into atomic knowledge:

$$\{K_1, K_2, \ldots, K_m\} = Decompose(D)$$

Compute similarity between query and each knowledge item:

$$f(Q, K_i) = \langle Enc(Q), Enc(K_i) \rangle$$

Select top-$k$ knowledge items by similarity:

$$D' = \{K_1', K_2', \ldots, K_k'\}$$

**Step 2: Hidden-State Probing for Conflict Detection**
**foreach** $K_i' \in D'$ **do**
    Obtain hidden representation from frozen model:

$$h_i = \mathcal{M}(K_i') \in \mathbb{R}^{d_M}$$

    Classify conflict via probing model $\mathcal{P}$:

$$y_i = \mathcal{P}(h_i) \in \{0, 1\}$$

    **if** $y_i = 1$ **then**
        Mark $K_i'$ with special tokens $\langle conflict \rangle K_i' \langle /conflict \rangle$
    **end**
**end**

**Step 3: Conflict-Aware Supervised Fine-Tuning (CA-SFT)**
**foreach** *conflicting knowledge item* $K_i'$ **do**
    Identify token positions $S = \{j \mid x_j \in T^{(i)}\}$
    Compute attention-guidance loss:

$$\mathcal{L}_{\text{Attn}} = \frac{1}{|P|} \sum_{(i,j) \in P} (1 - \alpha_{ij}), \quad P = \{(i, j) \mid i \geq j; j \in S\}$$

**end**
Combine with language modeling loss:

$$\mathcal{L}_{\text{Total}} = (1 - \lambda)\mathcal{L}_{\text{LM}} + \lambda \mathcal{L}_{\text{Attn}}$$

**Final Answer Generation**
Generate final answer $A$ using fine-tuned model $\mathcal{M}_{\text{CA-SFT}}$ conditioned on pruned and
annotated context $D'$.

**Algorithm 1:** CLEAR: Conflict-Localized and Enhanced Attention for RAG

---

> **Context Decomposition Prompt**
>
> Please breakdown the following context into independent atomic facts.
> Each fact must:
> Be written as a complete sentence.
> Use a specific entity name as the subject (avoid using vague subjects like "the" or "it").
> Preserve only one piece of information per sentence (no conjunctions that combine multiple facts).
> Stay faithful to the original text without adding extra interpretation.
>
> For example:
> Context: Christopher Nolan directed a 2006 film in which Ron Perkins' character plays the manager of a hotel.
> Facts:
> - Christopher Nolan directed a 2006 film.
> - Ron Perkins' character plays the manager of a hotel.
>
> Now please breakdown the following context:
> Context: {context}
> Facts:

Figure 5: Context decomposition prompt used in the Fine-Grained Knowledge Pruning module.

## A.2 IMPLEMENTATION DETAILS

**Detail of CLEAR.** For the implementation of CLEAR, we configure the experimental settings as follows. In the **Fine-Grained Knowledge Pruning** module, we employ gpt-3.5-turbo to decompose the retrieved context into fine-grained knowledge using the prompt template illustrated in Figure 5. We then compute semantic similarity among the decomposed knowledge with all-MiniLM-L6-v2 and retain the top-10 most relevant knowledge item.

In the **Hidden-State Probing for Conflict Detection** module, the selected knowledge items are fed into the model, from which we extract hidden states of the decoder. These representations are passed to a trained MLP-based probe for binary classification. The probe consists of three fully connected layers with ReLU activation, followed by a sigmoid normalization. For training, we sample 1,000 instances with a learning rate of 0.001 and train the probe for 10 epochs.

For the **Conflict-Aware Fine-Tuning** module, we set the weighting hyperparameter $\lambda = 0.1$. On the ConFiQA dataset, we allocate 13,500 instances for training (with 4,500 samples each from the MC, MR, and QA subsets), while the remaining data are reserved for evaluation. We fine-tune the model using LoRA, where the rank $r$ is set to 16, the scaling factor $\alpha$ to 16, and the learning rate to $3 \times 10^{-5}$, training for a total of 5 epochs. Finally, during inference, we set the temperature parameter to 0 to ensure reproducibility of results.

**Detail of Baseline.** For all baselines reported in the main experiments, we adopt a sampling temperature of 0 and a maximum generation length of 128 tokens. For CAD, we set the hyperparameter $\alpha = 0.9$. For all prompt-based methods, we directly employ the prompt templates provided in the original papers. For all training-based methods, we use the same training data as CLEAR, sampled from ConFiQA. Specifically, for Context-DPO, we apply the same LoRA configuration during training. For CANOE, we follow the original training setup and perform full-parameter fine-tuning on four NVIDIA A100 GPUs.

**Detail of Ablation Study.** For the **w/o Knowledge Pruning** variant, we partition the input context directly into sentences and subsequently apply the conflict detection module to determine whether each sentence conflicts with the model's parametric knowledge. For the **w/o Conflict Detection** variant, we fine-tune the model using the decomposed knowledge directly. Since conflicting knowledge is not explicitly identified, only the loss term $\mathcal{L}_{\text{LM}}$ is active during CA-SFT fine-tuning. For the **w/o CA-SFT** variant, we remove the $\mathcal{L}_{\text{Attn}}$ term, which reduces the training objective to standard SFT without attention-level supervision.

Table 4: Supplementary experimental results on additional model architectures.

| Method | FaithEval | | ConFiQA (MC) | | ConFiQA (MR) | | ConFiQA (QA) | | SQuAD | |
|---|---|---|---|---|---|---|---|---|---|---|
| | F1 | EM | F1 | EM | F1 | EM | F1 | EM | F1 | EM |
| **LLaMA-2-7B-Chat-HF** | | | | | | | | | | |
| Context-DPO | 63.2 | 50.7 | 57.9 | 32.0 | 58.5 | 32.7 | 73.7 | 64.7 | 62.4 | 41.8 |
| CANOE | **70.6** | 52.3 | 73.9 | **70.2** | 75.2 | 72.6 | 74.3 | 72.7 | 63.2 | 45.6 |
| CLEAR | 68.3 | **54.4** | **79.1** | 69.7 | **80.2** | **77.0** | **86.1** | **81.7** | **65.4** | **52.1** |
| **Qwen2.5-7B-Instruct** | | | | | | | | | | |
| Context-DPO | 65.1 | 50.2 | 62.7 | 53.7 | 71.1 | 58.8 | 75.0 | 66.3 | 55.2 | 36.4 |
| CANOE | **68.1** | **53.9** | 68.7 | 61.1 | 71.7 | 67.8 | 70.6 | 66.9 | 59.4 | 41.3 |
| CLEAR | 63.5 | 48.9 | **88.8** | **86.2** | **89.6** | **86.2** | **94.3** | **91.5** | **61.6** | **46.2** |

Table 5: Accuracy and Attention Weight across different $\alpha$ values for three models.

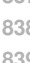

| $\alpha$ | LLaMA-3.1-8B-Instruct | | Qwen3-8B | | Mistral-7B-v0.3 | |
|---|---|---|---|---|---|---|
| | Accuracy | Attention | Accuracy | Attention | Accuracy | Attention |
| 0.0 | 0.552 | 0.020 | 0.512 | 0.115 | 0.631 | 0.070 |
| 0.1 | 0.644 | 0.105 | 0.604 | 0.022 | 0.663 | 0.193 |
| 0.2 | 0.632 | 0.188 | 0.573 | 0.195 | 0.598 | 0.203 |
| 0.3 | 0.635 | 0.231 | 0.639 | 0.283 | 0.559 | 0.211 |
| 0.4 | 0.554 | 0.331 | 0.495 | 0.415 | 0.611 | 0.314 |
| 0.5 | 0.482 | 0.442 | 0.443 | 0.390 | 0.538 | 0.463 |
| 0.6 | 0.333 | 0.464 | 0.430 | 0.381 | 0.289 | 0.543 |
| 0.7 | 0.201 | 0.483 | 0.153 | 0.474 | 0.171 | 0.549 |
| 0.8 | 0.214 | 0.481 | 0.211 | 0.444 | 0.117 | 0.459 |
| 0.9 | 0.210 | 0.464 | 0.207 | 0.457 | 0.194 | 0.538 |

## B  ADDITIONAL EXPERIMENT

### B.1  ADDITIONAL MODEL ARCHITECTURE FOR MAIN EXPERIMENT

Table 4 presents supplementary results on two additional model architectures, **LLaMA-2-7B-Chat-HF** and **Qwen2.5-7B-Instruct**, evaluated across multiple benchmarks. Consistent with the main findings, CLEAR demonstrates notable improvements over both Context-DPO and CANOE, particularly on conflict-sensitive datasets such as ConFiQA and FaithEval. For LLaMA-2-7B-Chat-HF, CLEAR achieves the highest scores on most ConFiQA variants, while also maintaining competitive performance on FaithEval and SQuAD.

On Qwen2.5-7B-Instruct, the advantage of CLEAR becomes even more pronounced: it consistently outperforms both baselines across all ConFiQA settings, with substantial gains in F1 and EM. Although CANOE occasionally remains competitive on less conflict-intensive benchmarks, CLEAR shows strong generalization in resolving conflicting knowledge. These results confirm that the effectiveness of CLEAR extends beyond a single backbone, underscoring its robustness across different instruction-tuned LLMs.

### B.2  SUPPLEMENTARY EXPERIMENTAL RESULTS ON ATTENTION ANALYSIS

Table 5 reports the detailed numerical results corresponding to Figure 4, including both the model accuracy and the attention weight assigned to conflicting knowledge across different values of $\alpha$ for LLaMA-3.1-8B-Instruct, Qwen3-8B, and Mistral-7B-v0.3. Consistent with the trends shown in the figure, attention weights increase steadily with larger $\alpha$, saturating around $\alpha = 0.5$. In contrast, accuracy peaks within a smaller range of $\alpha$ (0.1–0.3) and then declines as $\alpha$ continues to grow. These results highlight that while higher $\alpha$ values encourage stronger focus on conflicting knowledge, this emphasis can come at the cost of overall performance. The tabulated results thus provide a more fine-grained view of the trade-off between model attention allocation and accuracy under varying $\alpha$ values.

Table 6: Case Study. This table displays the knowledge extracted from the context and the results of identifying knowledge conflicts. Based on the conflicting knowledge, the model can correctly answer questions (even when the golden answer is counterfactual).

| Question | A group of engineers wanted to know how different building designs would respond during an earthquake. They made several models of buildings and tested each for its ability to withstand earthquake conditions. Which will most likely result from testing different building designs? |
|---|---|
| Context | Seismic testing of building models is crucial for understanding how structures will behave during earthquakes. Engineers approach these tests with a myriad of designs, each aiming to improve certain aspects of building performance, such as safety, aesthetic appeal, and construction speed... |
| Knowledge Extracted | (1) Seismic testing of building models is crucial for understanding structural behavior during earthquakes. 
 (2) Engineers approach tests with a myriad of designs aiming to improve safety, aesthetic appeal, and construction speed. 
 (3) $\langle conflict \rangle$ **Implementation of efficient techniques can enhance building times by up to 30%.** $\langle /conflict \rangle$ 
 (4) Seismic testing aligns efficiency with safety in contemporary civil engineering practices. 
 (5) $\langle conflict \rangle$ **Speed of construction is a dominant benefit of testing building designs under earthquake simulation conditions.** $\langle /conflict \rangle$ 
 (6) Optimization of construction speed guarantees resilience and rapid realization of new buildings through continued innovation and testing. 
 ... |
| Model Answer | Buildings will be built faster. |

## C   CASE STUDY

In this section, we present a case study to further illustrate how our proposed framework CLEAR enforces contextual faithfulness under knowledge conflicts. We conduct the analysis on the Faitheval dataset using the `LLaMA-3.1-8B-Instruct` model, and the results are shown in Table 6. CLEAR first decomposes the retrieved context into fine-grained knowledge, followed by filtering and conflict detection. As indicated in the table, the context explicitly states that construction speed is the dominant benefit of seismic testing, whereas the model's prior knowledge typically associates seismic testing with structural safety. Through our conflict detection probe, CLEAR successfully identifies such conflicts and, with the aid of CA-SFT, reinforces the model's attention to the conflicting knowledge (3) and (5). As a result, CLEAR generates the correct answer, *"Buildings will be built faster,"* which faithfully reflects the contextual evidence rather than relying on the model's internal knowledge. This case study highlights the effectiveness of our framework in ensuring contextual faithfulness in scenarios involving knowledge conflicts.

## D   LIMITATIONS

While CLEAR demonstrates strong improvements in textual RAG scenarios, its applicability to multimodal RAG systems remains limited. The current framework is designed around sentence-level textual decomposition and hidden-state probing, which are not directly transferable to modalities such as images, audio, or structured data. In multimodal contexts, knowledge conflicts may manifest in non-textual representations, requiring new strategies for knowledge decomposition, conflict detection, and attention guidance. Extending CLEAR to handle heterogeneous modalities would thus require substantial redesign of its probing mechanism and fine-tuning objectives, which we leave as an important direction for future research.

## E   RELATED WORK

In this appendix, we provide an extended review of related work on RAG, contextual faithfulness, and knowledge conflict, complementing the concise overview in Section 5.

**Retrieval-Augmented Generation**   RAG has become a cornerstone paradigm for improving the factual reliability and adaptability of LLMs by explicitly integrating external information during the generation process. Early contributions such as REALM (Guu et al., 2020c) and RAG (Lewis et al., 2020) pioneered the idea of end-to-end frameworks in which a retriever component selects relevant passages from large-scale corpora, which are then consumed by a generator to produce responses grounded in retrieved evidence. This framework demonstrated clear advantages over purely parametric models, particularly in tasks requiring factual precision or knowledge of recent events.

Following these foundational works, the research community has proposed a series of improvements targeting both the retriever and generator components. For retrieval, dense retrieval methods (Karpukhin et al., 2020; Izacard et al., 2023) introduced learned embeddings that outperform traditional sparse methods (e.g., BM25) in capturing semantic relevance. Subsequent refinements incorporated multi-vector representations (Santhanam et al., 2021), passage reranking (Nogueira & Cho, 2019), and adaptive retrieval strategies (Sun et al., 2022), where the retrieval budget is dynamically allocated based on the complexity of the query or the uncertainty of the model's predictions.

On the generator side, works have explored how to more effectively incorporate retrieved passages during decoding. FiD (Fusion-in-Decoder) (Izacard & Grave, 2020) demonstrated the effectiveness of late-fusion mechanisms, where a Transformer decoder attends jointly over multiple retrieved documents. Later works extended this paradigm with hierarchical fusion (Ram et al., 2023), sparse attention mechanisms (Shuster et al., 2022), and multi-hop retrieval pipelines (Xu et al., 2023). Hybrid models such as RePlug (Shi et al., 2023b) and Retro (Borgeaud et al., 2022) further integrated retrieval into pretraining or finetuning pipelines, blending parametric and non-parametric memories to achieve both scalability and factual accuracy. More recently, adaptive frameworks (Chen et al., 2024) proposed fine-grained controls over how retrieval signals are weighted depending on task type, query ambiguity, or user intent.

In addition to architectural innovations, researchers have also investigated the evaluation and efficiency of RAG systems. Benchmarks such as KILT (Petroni et al., 2020) and ELI5 (Fan et al., 2019) standardized evaluation across knowledge-intensive tasks, while efficiency-focused studies (Guu et al., 2020b) highlighted the trade-off between retrieval accuracy, latency, and resource consumption.

**Contextual Faithfulness**   Contextual faithfulness, defined as the degree to which model outputs remain consistent with retrieved or provided context, has emerged as a central concern in RAG research. Without explicit mechanisms to enforce faithfulness, models may hallucinate, overgeneralize, or generate outputs inconsistent with retrieved passages.

Prompt-based methods were among the earliest to address this challenge. Self-RAG (Asai et al., 2023) introduced self-reflection mechanisms, where models generate justifications for retrieved content and use these to re-ground their outputs. Template-based prompting approaches (Ying et al., 2024) designed structured query-response formats to encourage explicit grounding, though such methods often struggle with generalization across tasks.

Decoding-based approaches tackle faithfulness by modifying the generation process itself. Contrastive Decoding (Yuan et al., 2024) and Context-Aware Decoding (CAD) (Shi et al., 2023a) explicitly re-weight token probabilities during beam search to favor outputs aligned with retrieved context. Similarly, likelihood re-ranking techniques (Zhang et al., 2024) compare candidate responses against retrieved evidence to penalize hallucinations. These approaches maintain the flexibility of generation while reducing unfaithful responses.

Reinforcement learning (RL) has also been extensively applied to enhance contextual faithfulness. CANOE (Si et al., 2025) integrates reward models that explicitly score the grounding of responses in retrieved passages. Context-DPO (Bi et al., 2024a) extends direct preference optimization to context-aware settings, allowing LLMs to directly learn from pairwise comparisons of faithful versus unfaithful outputs. Such RL-based frameworks emphasize end-to-end optimization, reducing reliance on handcrafted prompts or decoding heuristics.

Beyond methodological innovations, recent surveys (Zhou et al., 2023b; Ji et al., 2023) highlight persistent challenges in faithfulness evaluation. Automatic metrics such as factual consistency (Thorne et al., 2018) or entailment-based scores (Falke et al., 2019; Guo et al., 2023) provide useful proxies

but often fail to capture nuanced inconsistencies or omissions. Consequently, many works advocate for human-in-the-loop evaluation frameworks to assess contextual grounding at scale.

**Knowledge Conflict**    Knowledge conflict arises when the retrieved evidence contradicts either the model's internal parametric memory or other retrieved documents, creating ambiguity in determining which knowledge to trust. This problem is particularly acute in dynamic knowledge environments, where information evolves over time or when sources exhibit bias or factual inconsistency.

A growing body of work has investigated mechanisms to detect, represent, and resolve knowledge conflicts. Astute RAG (Wang et al., 2025a) introduces a source-aware retrieval module, leveraging reliability estimation to assess which sources are more trustworthy in the face of contradictions. FaithfulRAG (Zhang et al., 2025b) explicitly models fact-level conflicts, decomposing retrieved evidence into atomic claims and guiding the generation process through a self-thinking phase that resolves inconsistencies.

Alternative approaches focus on information-theoretic principles. Swin-VIB (Wang et al., 2025b), for example, applies a variational information bottleneck to modulate the trade-off between fidelity to retrieved evidence and reliance on internal knowledge, thereby accommodating conflicts in a principled manner. Other works (Xu et al., 2024b) propose categorizing conflicts into types—such as temporal drift, factual contradiction, or perspective variance—and tailoring resolution strategies accordingly.

Recent research also extends conflict resolution beyond the text domain. Multimodal RAG systems (Gao et al., 2023; Xu et al., 2024c) face analogous challenges, as retrieved visual or audio evidence may not align with textual outputs. This motivates broader frameworks for consistency checking across modalities. Furthermore, evaluation efforts (Xu et al., 2024b) emphasize the need for standardized benchmarks that explicitly include conflict scenarios, enabling more systematic analysis of models' conflict-handling behaviors.

In summary, while significant progress has been made, knowledge conflict remains an open problem. Robust handling of contradictory information is critical not only for improving factual accuracy but also for building user trust in RAG-based systems deployed in real-world applications.

## F    THE USE OF LARGE LANGUAGE MODELS

In preparing this paper, we made limited use of Large Language Models (LLMs). Specifically, LLMs were employed for two purposes: (i) to aid in polishing the writing by improving grammar, readability, and clarity without altering the scientific content, and (ii) to assist in retrieval and discovery tasks, such as identifying and organizing related work. No LLMs were used for generating novel research ideas, designing experiments, or analyzing results. All conceptual and technical contributions presented in this paper are the sole work of the authors.

