# OpenReview forum: "Probing Latent Knowledge Conflict for Faithful Retrieval-Augmented Generation"
_ICLR.cc/2026/Conference — ICLR 2026 Conference Withdrawn Submission_

### Official Review · Reviewer_aJi6 · 2025-10-15

**Soundness:** 2
**Presentation:** 2
**Contribution:** 1
**Rating:** 2
**Confidence:** 4

**Summary:**

This paper proposes CLEAR, a framework to improve contextual faithfulness in RAG by: (1) decomposing context into sentence-level knowledge and filtering irrelevant items, (2) using hidden-state probing to detect conflicts with parametric memory, and (3) fine-tuning with attention-guidance loss to emphasize conflicting knowledge. The authors claim three novel insights from probing analysis: hierarchical knowledge integration, detectable latent conflict signals, and amplification of irrelevant aligned context. Experiments on ConFiQA, FaithEval, and SQuAD show 3-10% improvements over baselines.

**Strengths:**

1. The problem of internal-external knowledge conflict and detection in LLMs is interesting
2.  The paper achieves SOTA results across all settings, with particularly strong gains on conflict-heavy scenarios (e.g., ConFiQA-MC: +8-10% F1).

**Weaknesses:**

1.  The paper's most significant weakness is its failure to acknowledge and differentiate from highly relevant prior work. Specifically, existing studies [1] have already conducted thorough investigations of knowledge probing for internal-external conflicts and reported similar mechanistic insights. However, this paper neither cites these works nor compares against them as baselines, nor articulates what distinguishes its contribution from existing mechanistic analyses. Instead, it overstates the novelty of the "three key insights" by presenting them as original discoveries. This omission makes it impossible to assess the paper's true incremental contribution—if the core insights have been previously reported, the contribution may reduce to engineering improvements rather than fundamental discoveries. The authors must thoroughly review and cite relevant prior work on mechanistic analysis of knowledge conflicts, clearly differentiate their contributions, and reframe their novelty claims appropriately.

2. Probing generalization unclear: Probe trained on MQuAKE (knowledge editing) but tested on different conflict types. The assumption that "edited knowledge = conflicting knowledge" may not hold across domains. No analysis of what the probe actually learns or whether it captures spurious patterns.

3. Dependency on closed-source LLM: Knowledge decomposition relies on GPT-4o with no ablation of alternatives. This creates reproducibility concerns and cost barriers. Why not use open-source models or rule-based methods?

[1]. Towards Knowledge Checking in Retrieval-augmented Generation: A Representation Perspective  https://arxiv.org/abs/2411.14572

**Questions:**

1. How does your work differ from【1】 and other mechanistic analyses of knowledge conflicts? What is truly novel beyond engineering?
2. Can you provide ablations with open-source decomposition methods?
3. What does the probe actually learn? Feature attribution analysis?
4. How does CLEAR perform on tasks beyond extractive QA?
5. What are the computational costs compared to baselines?
6. Statistical significance of improvements?

[1]. Towards Knowledge Checking in Retrieval-augmented Generation: A Representation Perspective  https://arxiv.org/abs/2411.14572

---

### Official Review · Reviewer_cFHz · 2025-10-30

**Soundness:** 2
**Presentation:** 2
**Contribution:** 1
**Rating:** 2
**Confidence:** 5

**Summary:**

This paper tackles the contextual faithfulness problem in Retrieval-Augmented Generation (RAG) with large language models (LLMs). Using probing-based analysis of hidden states, the authors find that (1) LLMs integrate knowledge hierarchically, with failures emerging at the sentence level; (2) conflicting knowledge produces detectable latent signals; and (3) models overemphasize irrelevant context consistent with their parameters. Based on these insights, they propose CLEAR, a framework that decomposes context into sentences, probes hidden states to locate conflicts, and applies conflict-aware fine-tuning with an attention-guidance loss. Experiments across multiple benchmarks demonstrate the effectiveness of the proposed method.

**Strengths:**

- The idea of identifying conflict tokens is interesting and promising.

**Weaknesses:**

- The paper substantially overclaims novelty. It positions itself as the first mechanistic analysis of knowledge conflict, overlooking several prior works in this direction [1–3]. Moreover, Sections 2.1 and 2.2 largely reproduce findings already established in [4–6], offering limited new insight. The proposed method itself builds directly on these prior discoveries rather than introducing a genuinely novel perspective.
- Compared with [1–3], the probing analysis is relatively superficial and lacks depth in mechanistic interpretation. Consequently, the resulting method (Figure 3) becomes unnecessarily complex and tuning-heavy, which contradicts the paper’s mechanistic framing.
- Section 2.2 is poorly explained. Despite claiming hierarchical analysis, no actual layer-wise or structural examination is presented in the main paper.
- The experimental setup omits stronger recent baselines, particularly decoding-based and intervention-based methods [2,3,7–10]. Even if CLEAR marginally outperforms older baselines like CAD or COIECD, the need for extensive additional fine-tuning makes the comparison less meaningful.

[1] Characterizing mechanisms for factual recall in language models. ACL'23

[2] Cutting off the head ends the conflict: A mechanism for interpreting and mitigating knowledge conflicts in language models. ACL'24

[3] Taming Knowledge Conflicts in Language Models. ICML'25

[4] Adaptive Chameleon or Stubborn Sloth: Revealing the Behavior of Large Language Models in Knowledge Conflicts. ICLR'24

[5] The Power of Noise: Redefining Retrieval for RAG Systems. SIGIR'24

[6] The Geometry of Truth: Emergent Linear Structure in LLM Representations of True/False Datasets. COLM'24

[7] Sled: Self logits evolution decoding for improving factuality in large language models. NeurIPS'24

[8] Dola: Decoding by contrasting layers improves factuality in large language models. ICLR'24

[9] Active Layer-Contrastive Decoding Reduces Hallucination in Large Language Model Generation. EMNLP'25

[10] AdaCAD: Adaptively Decoding to Balance Conflicts between Contextual and Parametric Knowledge. NACCL'25

**Questions:**

- Do the authors need to train different probes for different models? What is the exact extra training runtime/FLOPS/resources required by CLEAR?
- In line 152, the authors state that “lower layers of LLMs capture token-level information, while deeper layers integrate sentence-level and passage-level semantics.” However, no quantitative or visual evidence is provided in the main paper. Could the authors clarify what empirical analysis supports this claim (e.g., layer-wise probing, representation similarity, or attention structure)?
- A recent work [1] demonstrates that fine-tuning on knowledge conflict pairs could lead to adversarial effects. How does CLEAR address or mitigate such potential adversarial effects when fine-tuning on conflict-annotated data?
- How can you measure the accuracy (e.g. generalization) of the trained probe?

[1] Context-Parametric Inversion: Why Instruction Finetuning Can Worsen Context Reliance. ICLR'25

---

### Official Review · Reviewer_RcND · 2025-11-01

**Soundness:** 2
**Presentation:** 3
**Contribution:** 2
**Rating:** 2
**Confidence:** 4

**Summary:**

This paper investigates contextual unfaithfulness in Retrieval-Augmented Generation (RAG) systems, particularly in scenarios involving knowledge conflicts. The authors claim to identify a "latent conflict signal" at the sentence level by probing hidden states. Based on this, they propose CLEAR, a framework that 1) decomposes and prunes context into atomic facts, 2) uses a probing model to detect conflicting knowledge, and 3) fine-tunes the model with an attention-guidance loss to prioritize the conflicting evidence and improve faithfulness.

**Strengths:**

The paper addresses a critical problem in RAG systems: the lack of contextual faithfulness, particularly in the challenging scenario where retrieved evidence conflicts with the model's internal parametric knowledge. The idea of allocating greater attention to conflicting knowledge is interesting.

**Weaknesses:**

- The preliminary study in Section 2.1, which examines LLM performance against irrelevant and counterfactual context, is presented as a novel exploration. This is a well-studied area in the literature (Longpre et al., 2021; Xie et al., 2024), and the paper fails to discuss how this study differs from or builds upon prior work. This risks overclaiming the novelty of these initial findings.
- The first stage of the CLEAR framework, "Fine-Grained Knowledge Pruning," requires an external call to a frontier model (GPT-4o or gpt-3.5-turbo, inconsistent between main text and appendix) for decomposition. First, the step is costly and must be performed during both training and inference. Second, using such a powerful external model acts as a confounding variable. It is unclear how much of the final performance improvement is attributable to the proposed probing and fine-tuning mechanism versus the advanced reasoning and decomposition capabilities of the external model.
- The "Conflict-Aware Fine-Tuning" module directly regularizes the model's attention distribution using an auxiliary loss. This approach can be problematic, as LLM attention patterns are often non-intuitive (e.g., attention sinks) and may not be a reliable proxy for information flow or importance. This is implied in the results with larger $\alpha$. Moreover, the paper omits critical implementation details, such as which specific layers or attention heads are being supervised.
- The selection of datasets is limited to mainly three sources.
- The paper's literature review is incomplete and misses several relevant works for resolving knowledge conflicts in RAG.

Reference

- ACL 2024, Truth-Aware Context Selection: Mitigating Hallucinations of Large Language Models Being Misled by Untruthful Contexts
- ICLR 2025, To Trust or Not to Trust? Enhancing Large Language Models' Situated Faithfulness to External Contexts
- EMNLP 2021, Entity-Based Knowledge Conflicts in Question Answering

**Questions:**

- How is the t-SNE visualizations in Figure 2 generated?
- The "Fine-Grained Knowledge Pruning" step requires a call to an external LLM, which is described as GPT-4o in line 239, and GPT-3.5-turbo in line 781, why the discrepancy? Can the model perform such composition themselves?

---

### Official Review · Reviewer_szNc · 2025-11-01

**Soundness:** 2
**Presentation:** 3
**Contribution:** 2
**Rating:** 4
**Confidence:** 3

**Summary:**

The paper investigates internal mechanisms behind contextual unfaithfulness in RAG, finding that (i) knowledge is integrated hierarchically (token→sentence→passage), (ii) latent conflict signals emerge at the sentence level before decoding errors, and (iii) models amplify irrelevant context when it aligns with parametric memory. Based on these findings, the authors propose CLEAR, comprising (1) fine-grained sentence-level knowledge pruning, (2) hidden-state probing (MLP) to localize conflicts, and (3) conflict-aware fine-tuning with an attention guidance loss. CLEAR achieves SOTA across FaithEval, ConFiQA (MC/MR/QA), and SQuAD over multiple backbones.

**Strengths:**

- Shifts from external interventions to internal representation analysis to detect conflicts. The t-SNE evidence that aligned vs. conflicting knowledge form separable clusters across models is compelling.

- Converts probing signals into attention guidance during fine-tuning, yielding robust gains under multiple conflict types and models

- Likely to influence RAG faithfulness research, bridging interpretability signals and training objectives.

**Weaknesses:**

- The pipeline adds pruning, probing, and CA-SFT steps; the paper lacks a runtime/throughput analysis for training and inference.

- Knowledge pruning via semantic similarity could prune subtle yet necessary evidence in multi-hop or low-similarity settings (the paper mitigates this partly with ConFiQA MR/MC, but more analysis would help)

- While probing-based detection is effective, it is trained separately from the main model, which might limit joint optimization of the reasoning pipeline.

**Questions:**

- What is the end-to-end overhead (per query) for decomposition, similarity filtering, probing, and CA-SFT inference?

- How does the conflict detector trained on MQuAKE perform on conflicts not involving entity edits (e.g., reasoning-chain contradictions as in FaithEval)?

- Can you share false-positive/negative cases for the probe and attention heatmaps showing how CA-SFT shifts focus?

---

### Note · Authors · 2025-11-25

I have read and agree with the venue's withdrawal policy on behalf of myself and my co-authors.